# Viral Metagenomics for the Identification of Emerging Infections in Clinical Samples with Inconclusive Dengue, Zika, and Chikungunya Viral Amplification

**DOI:** 10.3390/v14091933

**Published:** 2022-08-31

**Authors:** Juliana Vanessa Cavalcante Souza, Hazerral de Oliveira Santos, Anderson Brandão Leite, Marta Giovanetti, Rafael dos Santos Bezerra, Eneas de Carvalho, Jardelina de Souza Todão Bernardino, Vincent Louis Viala, Rodrigo Haddad, Massimo Ciccozzi, Luiz Carlos Junior Alcantara, Sandra Coccuzzo Sampaio, Dimas Tadeu Covas, Simone Kashima, Maria Carolina Elias, Svetoslav Nanev Slavov

**Affiliations:** 1Central Laboratory of Public Health, Maceio 57036-860, Brazil; 2Laboratory of Pharmacology and Immunity, Institute of Biological and Health Sciences, Federal University of Alagoas, Maceio 57051-090, Brazil; 3Department of Science and Technology for Humans and the Environment, University of Campus Bio-Medico di Roma, 00185 Rome, Italy; 4Laboratory of Flaviviruses, Oswaldo Cruz Foundation, Rio de Janeiro 21040-900, Brazil; 5Blood Center of Ribeirao Preto, Faculty of Medicine of Ribeirao Preto, University of Sao Paulo, Rua Tenente Catão Roxo 2501, Ribeirao Preto, Sao Paulo 14051-140, Brazil; 6Department of Biotechnology (NuCEL), Butantan Institute, Sao Paulo 05503-900, Brazil; 7Faculty of Ceilandia, University of Brasilia, Federal District, Brasília 70904-970, Brazil; 8Epidemiology and Statistic Unit, University of Campus Bio-Medico di Roma, 00185 Rome, Italy

**Keywords:** metagenomics, virome, emerging arboviruses, RT-qPCR, genotypes, dengue, Chikungunya, Brazil

## Abstract

Viral metagenomics is increasingly being used for the identification of emerging and re-emerging viral pathogens in clinical samples with unknown etiology. The objective of this study was to shield light on the metavirome composition in clinical samples obtained from patients with clinical history compatible with an arboviral infection, but that presented inconclusive results when tested using RT-qPCR. The inconclusive amplification results might be an indication of the presence of an emerging arboviral agent that is inefficiently amplified by conventional PCR techniques. A total of eight serum samples with inconclusive amplification results for the routinely tested arboviruses—dengue (DENV), Zika (ZIKV), and Chikungunya (CHIKV) obtained during DENV and CHIKV outbreaks registered in the state of Alagoas, Northeast Brazil between July and August 2021—were submitted to metagenomic next-generation sequencing assay using NextSeq 2000 and bioinformatic pipeline for viral discovery. The performed bioinformatic analysis revealed the presence of two arboviruses: DENV type 2 (DENV-2) and CHIKV with a high genome coverage. Further, the metavirome of those samples revealed the presence of multiple commensal viruses apparently without clinical significance. The phylogenetic analysis demonstrated that the DENV-2 genome belonged to the Asian/American genotype and clustered with other Brazilian strains. The identified CHIKV genome was taxonomically assigned as ECSA genotype, which is circulating in Brazil. Together, our results reinforce the utility of metagenomics as a valuable tool for viral identification in samples with inconclusive arboviral amplification. Viral metagenomics is one of the most potent methods for the identification of emerging arboviruses.

## 1. Introduction

The arboviral infections caused by Dengue (DENV), Zika (ZIKV), and Chikungunya (CHIKV) viruses represent a global health burden owing to the large magnitude of the outbreaks; the severity of the clinical symptoms; and the expanding range of the transmitting vectors, the urban-dwelling mosquitos *Aedes aegypti* and *Aedes albopictus* [1]. The symptoms of these infections include fever, rash, myalgia, arthralgia, headache, lymphadenitis, and retro-orbital pain, making their differential diagnosis based solely on clinical findings and pre-established clinical algorithms impossible [2]. For this reason, RT-qPCR on viremic serum is the test of choice for viral detection and differentiation at the beginning of the symptoms. Additionally, early molecular diagnosis can be of great value for the prevention of clinical complications like dengue shock syndrome and severe joint sequels induced by persistent CHIKV infection.

Owing to the simultaneous circulation of DENV, ZIKV, and CHIKV in endemic areas, the best diagnostic option is the application of high sensitivity and specificity multiplex RT-qPCR for viral differentiation and confirmation [3]. Although highly efficient, the combination of different viral detection targets, primers, and probes in a single tube also has its disadvantages. Low viral load infections, presence of co-infections, and underrepresented viral genomes may lead to probe competition, poor detection limit, cross-reactivity between the tested viruses, and as a consequence to falsely positive or negative diagnostic reactions, leading to indeterminate results [4].

A further challenge is the presence of co-circulating emerging arboviruses causing infections with similar in clinical symptoms, but that are not targeted by the available routine multiplex PCR systems. The most powerful method for viral identification in such situations is the viral metagenomics that, because of its unbiased approach to pathogen identification, reveals the total viral abundance in suspect samples [5,6]. Therefore, viral metagenomics has increased the potential for pathogen diagnosis compared with amplification assays. For that reason, in this study, we applied total viral metagenomics on plasma samples obtained from patients with symptoms of arboviral disease, but for which the routine diagnosis showed an indeterminate amplification profile using the routine DENV, ZIKV, and CHIKV kit (BioManguinhos, Brazil). Such a situation perplexed us and we suspected that such peculiar amplification might be related to the presence of emerging viruses that may weakly amplify because of similarities in the genomic sequences. The application of viral metagenomics was also used to confirm the viral genotype of infecting virus by phylogenetic analysis by the assembly of complete genomes. After the evaluation of the total viral abundance by metagenomics, the viruses of interest were individually confirmed by alternative qPCR primer and probe sets in all tested samples.

## 2. Materials and Methods

### 2.1. Clinical Samples

A total of eight serum samples that showed inconclusive amplification result when tested using a multiplex RT-qPCR assay for DENV, ZIKV, and CHIKV detection (molecular ZDC kit, Bio-Manguinhos, Rio de Janeiro, Brazil) were submitted to viral metagenomics. The samples were obtained during a 30–32 epidemiological week period (July–August, 2021) from different municipalities of the state of Alagoas (Maceio, Maragogi, Andaia, and Corpupire; for more information, see the map in Figure 1A showing DENV or CHIKV outbreaks). The samples’ metadata and characteristics are shown in Table 1.

### 2.2. Next-Generation Sequencing

Individual serum samples were pre-treated with Turbo DNAse (ThermoFisher Scientific (Waltham, MA, USA)) for host/bacterial DNA removal and were further pooled. The pooling of samples is a standard procedure in the metagenomic analysis in order to reduce the cost of the sequencing and to test the largest amount of samples possible. The whole pool volume was extracted using the High Pure Viral Nucleic Acid Large Volume kit (Roche, Basel, Switzerland). The extracted nucleic acids were submitted to reverse transcription using the Superscript III First-Strand Synthesis System (ThermoFisher Scientific) and amplification was performed by the QuantiTect Whole Transcriptome Kit (QIAGEN, Hilden, Germany). The libraries were prepared using the DNA Prep Library Preparation Kit (Illumina (San Diego, CA, USA)). The pair-end sequencing of the dual-indexed libraries was performed by Illumina NextSeq 2000 sequencing platform using the NextSeq P3 flowcell (300 cycles) (Illumina), following the manufacturer’s instructions.

### 2.3. Bioinformatic and Phylogenetic Analyses

The obtained raw sequence data were submitted to quality control analysis using FastQC v. 0.11.8 software. Trimming and adapter removal was performed applying TrimGalore v.0.6.6 and Fastp v.0.23.1 to select sequences with the best quality and free of adapters. For metagenomic analysis, only reads with quality scores >30 were used. To infer the taxonomic classification of the virome, we used the Kraken2 v.2.0.8 program. Kraken2 was also used to subtract the human, bacterial, and parasitic reads, which were excluded from further analysis. “De novo” assembly was performed using SPAdes v.3.13.0 to generate viral contigs. Finally, to perform taxonomic classification based on protein identity, we applied Blastx as implied by the Diamond v.0.9.29 software. This pipeline was also applied for contaminant and artifact screening, which were removed after the BlastX step. The viral reads belonging to viruses infecting humans were manually curated and used for downstream analysis including phylogenetic studies.

Phylogenetic analysis was performed using a dataset of complete genomes obtained from the NCBI (National Center for Biotechnology Information, https://www.ncbi.nlm.nih.gov/ accessed on 10 August 2022). The alignment was performed using MAFFT v.7.429 and the reconstruction of the phylogenetic history of the identified viruses was performed using IQ-TREE v.18, applying the approximate maximum likelihood method with a statistical support of ultrafast bootstrap with 1000 replicates. The phylogenetic tree was visualized and edited in FigTree v.1.4.4 v. software.

### 2.4. Molecular Confirmation of Viral Agents

After the metagenomic identification of the viruses, the pool was opened and the viruses of interest were individually confirmed by individual qPCR tests in each sample. The viral confirmation was performed with alternative primer/probe systems compared to the routine diagnostic kit. The alternative DENV-2 and CHIKV molecular detection systems were already described in the literature [7,8]. RT-PCR was performed with 900 nm of each primer and 250 nm of probe using the GoTaq^®^ One-Step RT qPCR system (Promega). The amplification was carried out using the following protocol: initial steps of 45 °C for 25 min and 95 °C for 5 min, followed by 40 cycles consisting of denaturation at 95 °C for 15 s and 60 °C for 1 min in Quant Studio^TM^ 5 real-time PCR system (ThermoFisher). All of the procedures were performed in separate laboratory rooms and measures to prevent contamination were strictly followed.

## 3. Results

### 3.1. Metagenomic Analysis of the Samples

The NGS of the assembled pool generated a total number of 101,593,212 reads. Viral reads classified by Kraken2 represented 0.47% of the total number of reads obtained (478,390 viral reads). The generated Kraken2 output identified reads that belonged to arboviruses: DENV type 2 (69,091 reads) and CHIKV (372,026 reads) permitting the assembly of two complete genomes (92.2% of all obtained viral reads) (Figure 1B). Although the initial idea of the study was to only identify which viruses were present in the clinical samples, the presence of complete viral genomes allowed in-depth studies related to the molecular epidemiology of both viruses. The metagenomic analysis did not show the presence of ZIKV sequencing or other emerging arboviruses.

The metagenomic analysis also showed the presence of multiple commensal viruses that constitute a normal part of the human virome with most abundant species torque teno viruses (TTVs) of types 1, 3, 6, 8, 10–13, 18–20, 22, and 29. Human pegivirus-1 (HPgV-1) and human gemykibivirus-2 (HuGkV-2) were also identified at a low read level (Figure 1B). The total virus abundance of the tested pool is shown in Figure 1C.

### 3.2. Phylogenetic Analysis of Dengue and Chikungunya Viruses

The assembled genomes of DENV-2 and CHIKV were also analyzed using phylogenetic inference. The phylogenetic analysis of the DENV-2 genome showed that it belongs to the Asian/American genotype (Figure 2A). The obtained isolate was grouped in a monophyletic cluster with other Brazilian DENV-2 strains obtained from a large outbreak of this virus in Brazil during 2019–2020, mostly circulating in the state of São Paulo. The CHIKV phylogenetic reconstruction showed that our strain belonged to the ECSA genotype, which is the most prevalent in Brazil. Additionally, the strain clustered together with other strains circulating predominantly from the northeast part of Brazil, where our samples were also obtained (Figure 2B) and frequent CHIKV outbreaks occur.

### 3.3. Molecular Confirmation of the Arboviral Infections

We performed individual sample testing different from the routine RT-qPCR primer/probe sets for DENV and CHIKV detection. Our results showed that one of the eight samples was positive for DENV-2 with a cycle threshold (Ct) = 29.0 (sample 7, see Table 1) and two were CHIKV positive (samples 1 and 8) with a Ct = 25.0 and 26.0, respectively. During the individual amplification, we did not observe indeterminate results for both arboviral agents.

## 4. Discussion

The peculiar indeterminate amplification profile of routinely tested viruses might be related to poor annealing of the primers to emerging viral genomes with similar organization and, as a consequence, the presence of viruses that are not diagnosed by the applied routine diagnostic tests. This is an important hallmark of arboviruses, especially flaviviruses and alphaviruses, where the similarity between commonly presented genes, used widely in laboratory diagnoses, may lead to an indeterminate PCR profile, related to the presence of emerging arboviruses. Therefore, in this study, we applied metagenomics to clinical samples with indeterminate qPCR amplification obtained during arboviral outbreaks in a highly endemic area for CHIKV and DENV in order to shed light on the presence of co-circulating emerging arboviruses. The used bioinformatic pipeline showed the presence of two arboviruses of clinical interest: DENV-2 and CHIKV, with high genomic coverage enabling the performance of a phylogenetic analysis. Other emerging or novel viruses were not identified.

Viral metagenomics has been of great value for confirmation of emerging or unknown viral agents in clinical samples with undetermined diagnosis and for viral discovery [9,10]. The major advantages of metagenomics include high specificity of viral identification and the unbiased nature of viral detection, which enables full annotation of all viruses present in the tested sample [11,12]. Here, we identified two arboviruses with clinical interest (DENV-2 and CHIKV) and a wide range of commensal viruses including TTV species, human gemycircularviruses (an emerging commensal virus), and HPgV-1. On the other hand, the applied metagenomic analysis did not identify ZIKV reads and the positive results observed by qPCR were probably the result of non-specific amplification. Therefore, the currently used metagenomic and bioinformatic approaches are suitable for virus discovery and might be intensively applied in areas with a high risk of emergence of novel viruses, or where frequent outbreaks of known arboviral agents might occur.

Another great value of metagenomics, also observed in our study, was that it permitted not only the identification of viral agents, but also analyses of the molecular epidemiology of these agents. The obtained complete DENV-2 genome was classified as Asian/American genotype and was clustered with other Brazilian strains that caused an outbreak in the states of São Paulo and Rio de Janeiro in 2019–2020 [13]. We also obtained information for the molecular epidemiology of the identified CHIKV strain, which belonged to the ECSA genotype, which circulates intensively in northeast Brazil including the state of Alagoas [14] and was clustered with CHIKV strains from this Brazilian location. Therefore, the application of viral metagenomics provides valuable information, not only on the viral abundance or presence of emerging viruses, but also on the viral molecular epidemiology, which can influence infection control and public health measures [12].

We confirmed all identified arboviruses by alternative qPCR, identifying the positive samples. We are unaware of what exactly may have contributed to the divergence of the results obtained by the commercially available kit and the in-house primers and probes used. Multiple factors might have contributed to this variation. On one hand, we applied the in-house assays individually for each virus. The combination of multiple targets, primers, and probes in one reaction may lead to the formation of dimers, non-specific probe binding, and liberation of fluorescent signal, especially in the final stages of the reaction (the commercial protocol had five more cycles compared with the in-house protocol). Another possibility is the use of different target viral genes in the commercial and in-house assay used, which can lead to different levels of specificity and sensitivity. Finally, different laboratory practices and conditions may have contributed to these discrepant results.

In conclusion, we successfully applied viral metagenomics in clinical samples with indeterminate RT-qPCR results for the routinely tested arboviruses in search of emerging viral agents. The obtained bioinformatic results showed that the metagenomic approaches used were capable of identification with high precision the infecting agent and provided us with useful information for the molecular epidemiology. Nevertheless, viral metagenomics and bioinformatic analysis have a vast and diverse number of challenges that limit their routine application in the clinical practice, varying from validation, standardization, cost, turn-around time, and complex computer requirements. However, when the molecular methods fail to identify the etiological agent, and in cases of extreme emergencies (hemorrhagic diathesis or shock often caused by the arboviruses), viral metagenomics is the method of choice. Further improvements to the next-generation sequencing, especially in the input number of samples and reduction of the cost of reagents and the sequencing equipment, will shape the further application of metagenomics as a useful laboratory test in the clinical practice.

## Figures and Tables

**Figure 1 viruses-14-01933-f001:**
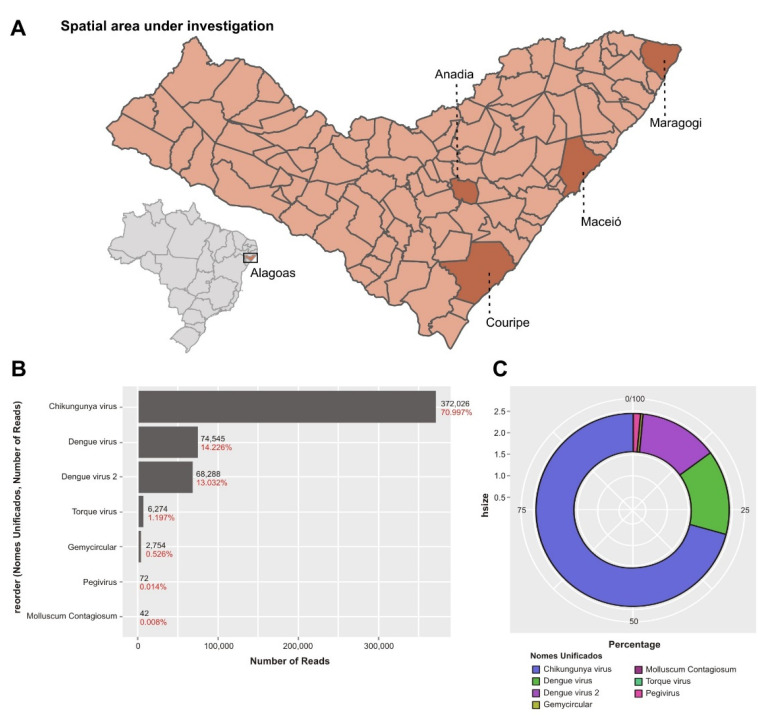
Area of investigation and metagenomic analysis of the analyzed pool of samples. (**A**) Map representing the spatial area under investigation and position of state of Alagoas in Brazil. (**B**) Bar-plots illustrating the relative abundance of viral reads for the most presented viral agents in the metavirome of the tested patients with predominance of Chikungunya and Dengue viral reads and other commensal viruses, forming part of the normal metavirome. (**C**) Circular representation of the relative viral abundance; Chikungunya and Dengue viruses are highly present.

**Figure 2 viruses-14-01933-f002:**
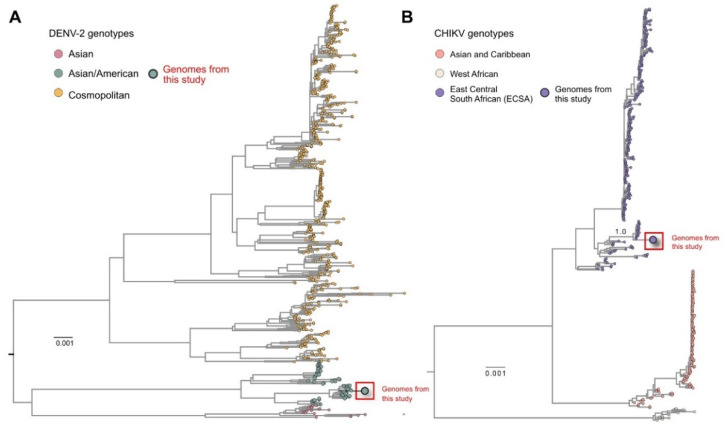
Phylogenetic analysis of the obtained complete Dengue and Chikungunya complete genomes. (**A**) Maximum likelihood tree showing that the new Dengue virus-2 genome sequences obtained in this study belonged to the Asian/American genotype and clustered together with other strains obtained from Brazil during the 2019–2020 DENV outbreak, especially strains that circulated in the São Paulo state. (**B**) Maximum likelihood tree analyzing complete Chikungunya genome sequence indicating that it belongs to the ECSA genotype, which is circulating in Brazil and especially the northeast, where the sample collection was performed.

**Table 1 viruses-14-01933-t001:** Characteristics of the tested samples and their amplification profile.

Sample ID	Type	Location	Symptom Onset	Dengue Virus ^1^	Zika Virus ^2^	Chikungunya ^3^	Dengue Type-2 ^4^	Chikungunya ^5^
1	serum	Maragogi	10 August 2021	Inconclusive	Undetectable	Detectable (Ct = 26.9)	Undetectable	Detectable (Ct = 25.8)
2	serum	Anadia	7 August 2021	Detectable (Ct = 28)	Detectable (Ct = 36)	Detectable (Ct = 31)	Undetectable	Undetectable
3	serum	Maceio	1 August 2021	Inconclusive	Undetectable	Detectable (Ct = 36.6)	Undetectable	Undetectable
4	serum	Maragogi	26 July 2021	Inconclusive	Undetectable	Detectable (Ct = 37.5)	Undetectable	Undetectable
5	serum	Coruripe	26 July 2021	Inconclusive	Undetectable	Detectable (Ct = 32.6)	Undetectable	Undetectable
6	serum	Maceio	10 August 2021	Detectable (Ct = 32)	Undetectable	Detectable (Ct = 35.4)	Undetectable	Undetectable
7	serum	Maceio	8 August 2021	Detectable (Ct = 18)	Detectable (Ct = 33.8)	Detectable (Ct = 30.9)	Detectable (Ct = 29.27)	Undetectable
8	serum	Maragogi	10 August 2021	Inconclusive	Undetectable	Detectable (Ct = 29.4)	Undetectable	Detectable (Ct = 26)

Legend: ^1, 2,^ and ^3^: Amplification profile obtained by the use of commercial kit for simultaneous detection of Dengue, Zika, and Chikungunya viruses (ZDC kit, Bio Manguinhos, Rio de Janeiro); ^4^: Dengue virus-2 testing by the detection system established by Johnson et al., 2005 (see [7]); ^5^: Chikungunya virus testing by the real-time PCR system established by Lanciotti et al., 2007 (see [8]).

## Data Availability

The generated sequences were deposited in GenBank under the following accession numbers: ON204004 and ON207129.

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
