# Peer review of "Viral Metagenomics for the Identification of Emerging Infections in Clinical Samples with Inconclusive Dengue, Zika, and Chikungunya Viral Amplification"

_viruses, 2022, doi:10.3390/v14091933_

Round 1

Reviewer 1 Report

The manuscript by Cavalcante Sousa et al. addresses the use of viral metagenomic sequencing to study clinical specimens with inconclusive PCR results. The authors examined 8 serum samples with inconclusive amplification results for their metaviromic composition. Although the results are interesting and highlight the utility of metagenomics as a valuable tool for clinical diagnosis, there are some important observations that remain to be addressed. T

Major comments:

1) The authors concluded that metagenomic sequencing is valuable for inconclusive results, but this was not clearly presented in the manuscript because it is not clear what was sequenced in each sample.

2) How many sequences did they obtain? Were they submitted to GenBank?

3) Why did the authors pool the clinical samples before isolating the nucleic acids? How can they assess coinfections if the samples were pooled?

4) It is not clear from the manuscript whether the confirmatory PCRs were performed with the same nucleic acids as the original PCR assays?

5) Was the nucleic acid extracted for NGS also tested using molecular techniques?

6) How do the authors comment on major discrepancies between the original and confirmatory tests?

7) How do the authors comment on the fact that no Zika virus was detected with NGS?

Minor comment:

1) The colours of the genomes obtained in this study in Figure 1 D and E are not visible.

2) The word "emerging" should be corrected in the title.

Author Response

REVIEWER #1

The manuscript by Cavalcante Sousa et al. addresses the use of viral metagenomic sequencing to study clinical specimens with inconclusive PCR results. The authors examined 8 serum samples with inconclusive amplification results for their metaviromic composition. Although the results are interesting and highlight the utility of metagenomics as a valuable tool for clinical diagnosis, there are some important observations that remain to be addressed.

We are grateful for the positive evaluation of Reviewer#1 regarding the presented manuscript. Thank you very much!

MAJOR COMMENTS

  • The authors concluded that metagenomic sequencing is valuable for inconclusive results, but this was not clearly presented in the manuscript because it is not clear what was sequenced in each sample.

We are grateful for the valuable comment of reviewer #1.

During an arboviral outbreak with a mixed circulation of Dengue (DENV) and Chikungunya viruses (CHIKV) in the Northeast Brazilian state of Alagoas (end of 2021), when applied the routine qPCR testing we observed high number of samples with inconclusive amplification results. These results were obtained using a commercially available kit provided from the Brazilian Ministry of Health that simultaneously amplifies three arboviruses with impact to public health i.e. DENV, Zika virus (ZIKV) and CHIKV. The presence of multiple indetermined results (mostly contributed to simultaneous presence of DENV, ZIKV and CHIKV positive results) puzzled us and we suspected that these samples may contain nucleic acids of emerging arbovirus agent that have similar genetic organization but is impossible to be amplified by the routine diagnostic tests giving this unspecific amplification profile. On the other hand, the question if these samples were in fact positive for the three viral infections was also plausible.

The most suitable approach to investigate if emerging viruses were present in these samples was the application of total viral metagenomics. This methodology by an unbiased amplification describes the abundance of all viruses that are present in the clinical samples. To verify initially which arboviruses were in fact present in these clinical samples, we applied the metagenomics on the pool assembled of all these samples, and after these we opened this pool and confirmed each virus individually by individual qPCR.

By applying metagenomics, we did not searched for specific viruses in each sample but for all the viral abundance. The assembled pool was submitted to all necessary steps for viral metagenomics including viral concentration, extraction of the whole volume of the pool with a large-volume extraction viral kit, reverse transcription using a random primers, isothermal amplification, library preparation and next generation sequencing. After the laboratory procedures we applied a bioinformatic pipeline for viral abundance based mainly on Kraken2 taxonomic classification and subsequent assembly of the reads into genomes and their phylogenetic analysis.

We are sorry that in the manuscript it was not clear that we performed a global metagenomics on the assembled pool and therefore, we added the following modifications:

Abstract:

Lines 30-31, Page 1: We added the following phrase: “… The inconclusive amplification results might be an indication for the presence of emerging arboviral agent that is inefficiently amplified by conventional PCR techniques. …”

Lines 41-44, Page 1: The conclusion of the abstract was modified “… Together our results reinforce the utility of metagenomics as valuable tool for viral identification on samples with inconclusive arboviral amplification. Viral metagenomics is one of the most potent methods for the identification of emerging arboviruses. …”

We added the following explanation in the “Introduction” section:

Lines 74-83, Page 2: “… For that reason, in this study we applied total viral metagenomics on plasma samples obtained from patients with symptoms of arboviral disease but for which the routine diagnosis showed indeterminate amplification profile using the routine DENV, ZIKV and CHIKV kit (BioManguinhos, Brazil). Such a situation perplexed us and we suspected that such peculiar amplification might be related to the presence of emerging viruses that may weakly amplify due to similarities in the genomic sequences. The application of viral metagenomics was also used to confirm the viral genotype of infecting virus by phylogenetic analysis by the assembly of complete genomes. After the evaluation of the total viral abundance by metagenomics, the viruses of interest were individually confirmed by alternative qPCR primer and probe sets in all tested samples. …”

2)  How many sequences did they obtain? Were they submitted to GenBank?

We are grateful for the valuable question of reviewer#1.

The overall sequencing of the pool (containing the 8 plasma samples) generated 372,026 sequence reads for Chikungunya virus and 143,833 reads belonging to Dengue-2. Other arboviruses were not identified. The obtained reads were sufficient to assemble two complete genomes of these viruses that were deposited in the GenBank under the following GenBank numbers ON204004 and ON207129. The objective of this study was to identify by metagenomics if in the samples there were emerging arboviruses and therefore, we did not perform individual sequencing of samples as we were unaware exactly which viruses were present. Our intention was not also to perform phylogenetic analysis but the presence of complete viral genomes naturally raised the question for the genotypes of the circulating viruses. The application of viral metagenomics revaled the viral abundance of the tested samples we were also able to find many commensal viruses with varying read number including many torque-teno viruses (TTV) 1, 12, 13, 16, 18, 22 and 29 and the gemycircularvirus SL1 which is an emerging virus with not well identified characteristics.

We added the following explanations in the manuscript text:

Line 143, Page 3: “… two complete genomes (92.2% of all obtained viral reads) …”

Lines 144-146, Page 3: “… Although, the initial idea of the study was to only identify which viruses were present in the clinical samples, the presence of complete viral genomes allowed in-depth studies related to the molecular epidemiology of both viruses. …”

Line 280, Page 7: We present the GenBank submission numbers of the two assembled sequences “…Data Availability Statement: The generated sequenced were deposited in the GenBank under the following accession numbers ON204004 and ON207129. …”

  • Why did the authors pool the clinical samples before isolating the nucleic acids? How can they assess coinfections if the samples were pooled?

We are grateful for the valuable comment of Reviewer#1.

The use of sample pools for next-generation sequencing is a standard procedure in the metagenomics to reduce the sequencing cost and to test more samples. By such a procedure we can have an overall vision of what is present in the samples and this gives further information for what viruses we will test the samples individually. Nevertheless, despite the pooling of samples, the total volume of the pool is submitted to viral concentration. Moreover, the total volume of the pool is extracted by a large volume extraction kit that permits minimal loss of samples and more exact evaluation of the viruses present in this type of material.

Considering the co-infections in the individual samples we adopted the following approach: (i) first we evaluated by metagenomic analysis the viruses that were present in the samples and (ii) the identified viruses were individually confirmed by individual alternative qPCR reactions in each sample. The CHIKV was tested using the molecular primer/probe set established by Lanciotti R et al., 2007 (Lanciotti et al., 2007) and the DENV was tested by primer system capable of detecting all circulating serotypes (Huhtamo et al., 2010). Using these tests we identified that two of the samples were positive for CHIKV RNA and one was positive for DENV-2 RNA. As we tested individually each sample, we did not observed co-infections.

The following modifications were added to the manuscript in order to become more clear:

Lines 95-97, Page 2: “… The pooling of samples is a standard procedure in the metagenomic analysis in order to reduce the cost of the sequencing and to test the largest amount of samples possible. …”

Lines 126-127, Page 3: “After the metagenomic identification of the viruses, the pool was opened and the viruses of interest were individually confirmed by individual qPCR tests in each sample. …”

  • It is not clear from the manuscript whether the confirmatory PCRs were performed with the same nucleic acids as the original PCR assays?

We are grateful to the reviewer for the valuable comment.

We opened pool that was metagenomically analyzed and knowing which viruses were present we tested their presence by individual qPCR reactions. The testing was performed by qPCR systems different from the commercially available. As commented above CHIKV RNA was tested using the molecular system that was established by Lanciotti et al., 2007. The presence of DENV RNA was tested by molecular system capable of detecting all DENV serotypes established by Huhtamo et al., 2010. Despite the fact that we did not identify ZIKV by the metagenomics, we tested its presence by the molecular system established by Lanciotti et al., 2008. As expected no positive results were obtained for this virus. We added the following information in the text:

Lines 128-130, Page 8: “… The viral confirmation was performed with alternative primer/probe systems com-pared to the routine diagnostic kit. The alternative DENV-2 and CHIKV molecular de-tection systems were already described in the literature [7,8]. …”

  • Was the nucleic acid extracted for NGS also tested using molecular techniques?

We are grateful for the valuable comment of reviewer 1.

We opened the pool that was composed of 8 plasma samples and all of them were were confirmed individually for the detected by metagenomic analysis viruses. The performed molecular reactions were also separate. ZIKV was also tested in these samples, but as expected all samples were negative.

  • How do the authors comment on major discrepancies between the original and confirmatory tests?

We are grateful to the reviewer for this valuable comment.

We are unaware of the reason that might have contributed to the different amplification profile of the results obtained by the commercially available kit in Brazil and the used in-house primer and probe sequences. We discuss some of the possible reasons below.

One of the possible causes includes the individual testing of the samples for each virus by the in-house kit. We performed separate testing of each sample which is time-consuming and not cost-effective but the specificity is better compared to the multiplex assays probably due to the detection of only one target. The presence of multiple primers and probes in the samples in combination with the short size of the amplified targets may lead to formation of dimers, non-specific probe annealing and liberation of fluorescent signal. This could explain why we observed amplification of the three viruses in some of the samples.

On the other hand, the differently applied assays may use different target genes which can also lead to different sensitivity and specificity of the reactions. If we suppose that the commercial test uses very similar or closely located viral genes for amplification of the three viruses, this may lead to non-specific amplification of the triple targets. We also believe that the higher number of cycles used in the commercially available kits may lead to final non-specific fluorescence and appearance of the cycle-threshold. Finally, different laboratory conditions may have contributed to the difference observed in the amplification of the viral targets by both assays.

We included the following explanation in the text in the “Discussion” section:

Lines 229-239, Pages 6,7: “… We confirmed all identified arboviruses by alternative qPCR identifying the positive samples. We are unaware of what exactly may have contributed to the divergence of the results obtained by the commercially available kit and the in-house used primers and probes. Multiple factors might have contributed to this variation. On one hand, we applied the in-house assays individually for each virus. The combination of multiple targets, primers and probes in one reaction may lead to formation of dimers, non-specific probe binding and liberation of fluorescent signal especially in the final stages of the reaction (the commercial protocol had 5 more cycles compared to the in-house protocol). Another possibility is the use of different target viral genes in the commercial and in-house used assay that can lead to different specificity and sensitivity. Finally different laboratory practices and conditions may have contributed to these discrepant results. …”

  • How do the authors comment on the fact that no Zika virus was detected with NGS?

We are grateful for the valuable comment of reviewer 1.

This is an important question that shows the objective of our study: to show the applicability of metagenomics for correct identification of the infecting viruses in cases of discordant, negative or positive PCR results. The applied viral metagenomics showed ZIKV amplification. In this respect, metagenomic analysis gives qualitative overview of all viral genomic material that is present of the samples and is more sensitive approach compared to the qPCR in that you can see directly the viral sequences. The PCR reaction is interpreted by its amplification curve and in some cases as discussed above the interpretation of the PCR results might be challenging due to non-specific amplification and especially when multiple probes, primers and similar targets are used. We agree that multiplex PCR reactions are cost-effective and very useful in the clinical practice. In this study we only show the applicability of the metagenomics in doubtful cases, for viral identification and how can both techniques complement each other. Moreover, the use of metagenomics for identification of novel viruses is undoubtful and in this study we show how using metagenomics led to correct identification of the infecting agents when the obtained results by real-time PCR were difficult to be interpreted. We added the following explanation in the manuscript text:

Lines 211-213, Page 6: “… On the other hand, the applied metagenomic analysis did not identify ZIKV reads and the observed positive results by qPCR were probably result of non-specific amplification. …”

MINOR COMMENT

1) The colours of the genomes obtained in this study in Figure 1 D and E are not visible.

We are grateful for the valuable suggestion of reviewer 1.

We modified the figure, thus making the identified virus genomes more visible. In that line, we split the figure presented in the initial version of the manuscript into to figures to make more visible the phylogenetic trees.

2) The word "emerging" should be corrected in the title.

We are very sorry for this involuntary error. The word was corrected in the title.

REFERENCES USED IN THIS REPLY

Asplund M, Kjartansdóttir KR, Mollerup S, Vinner L, Fridholm H, Herrera JAR, Friis-Nielsen J, Hansen TA, Jensen RH, Nielsen IB, Richter SR, Rey-Iglesia A, Matey-Hernandez ML, Alquezar-Planas DE, Olsen PVS, Sicheritz-Pontén T, Willerslev E, Lund O, Brunak S, Mourier T, Nielsen LP, Izarzugaza JMG, Hansen AJ. Contaminating viral sequences in high-throughput sequencing viromics: a linkage study of 700 sequencing libraries. Clin Microbiol Infect. 2019 Oct;25(10):1277-1285. doi: 10.1016/j.cmi.2019.04.028. Epub 2019 May 4. PMID: 31059795.

Huhtamo E, Hasu E, Uzcátegui NY, Erra E, Nikkari S, Kantele A, Vapalahti O, Piiparinen H. Early diagnosis of dengue in travelers: comparison of a novel real-time RT-PCR, NS1 antigen detection and serology. J Clin Virol. 2010 Jan;47(1):49-53. doi: 10.1016/j.jcv.2009.11.001. Epub 2009 Dec 5. PMID: 19963435.

Lanciotti RS, Kosoy OL, Laven JJ, Panella AJ, Velez JO, Lambert AJ, Campbell GL. Chikungunya virus in US travelers returning from India, 2006. Emerg Infect Dis. 2007 May;13(5):764-7. doi: 10.3201/eid1305.070015. PMID: 17553261; PMCID: PMC2738459.

Lanciotti RS, Kosoy OL, Laven JJ, Velez JO, Lambert AJ, Johnson AJ, Stanfield SM, Duffy MR. Genetic and serologic properties of Zika virus associated with an epidemic, Yap State, Micronesia, 2007. Emerg Infect Dis. 2008 Aug;14(8):1232-9. doi: 10.3201/eid1408.080287. PMID: 18680646; PMCID: PMC2600394.

Nunes Valença I, Silva-Pinto AC, Araújo da Silva Júnior W, Tadeu Covas D, Kashima S, Nanev Slavov S. Viral metagenomics in Brazilian multiply transfused patients with sickle cell disease as an indicator for blood transfusion safety. Transfus Clin Biol. 2020 Nov;27(4):237-242. doi: 10.1016/j.tracli.2020.07.001. Epub 2020 Aug 3. PMID: 32758666.

Reviewer 2 Report

The authors aim to describe the benefits of using sequencing and bioinformatics for the detection of arboviral infections. However, the authors fail to contextualize both in the methodology and results, the difference between sequencing and bioinformatics platforms for use in research vs in clinic. The advantage of utilizing sequencing workflows to detect and diagnose infectious diseases are numerous; however, these technologies have yet to become adopted due to a vast and diverse number of challenges ranging from validation, standardization, cost, turnaround time, etc. 

The methodology described by the authors is insufficient to determine the quality and applicability of the results presented. I strongly recommend to incorporate a quality system around the sequencing workflow and bioinformatics to determine detection limits, sensitivity, specificity, positive predictive value, negative predictive value for the viral targets of interest. Surely, after employing such framework, the impact of these results would increase substantially. 

I also strongly recommend extensive editing to improve on the spelling, grammar, and syntax text in the manuscript. The title has a glaring typographical error (Emrging -> Emerging).   

Author Response

REVIEWER #2

The authors aim to describe the benefits of using sequencing and bioinformatics for the detection of arboviral infections. However, the authors fail to contextualize both in the methodology and results, the difference between sequencing and bioinformatics platforms for use in research vs in clinic. The advantage of utilizing sequencing workflows to detect and diagnose infectious diseases are numerous; however, these technologies have yet to become adopted due to a vast and diverse number of challenges ranging from validation, standardization, cost, turnaround time, etc.

We are grateful for the comment of Reviewer #2.

In this study, the metagenomic application was directed to characterize emerging viruses in samples with peculiar amplification profile using bioinformatics. We supposed that samples with inconclusive amplification may contain not characterized viruses, which are co-circulating and weakly amplifying due to similarities in the viral genomes. The consequent application of the metagenomics showed that it not only identified the infecting viral agents and was able to taxonomically classify the viral genomes, but also directed the molecular amplification tests for further confirmation of these viruses in the individual samples.

We agree that the bioinformatic analysis and next-generation sequencing technologies might deal with many challenges that include beyond the mentioned by the reviewer, presence of viral contaminants, artifacts and commensal viruses that make the interpretation of the results relatively complex. Here also enter the question of the “dark matter”, which is represented by the non-classified sequences showing the inability of the currently used genomic classifiers to identify all of the present reads in the sequenced sample. In our opinion, the NGS, the bioinformatic analysis and the molecular detection methods must complement each other. In that line, the identification of viral reads by metagenomics must be always confirmed by molecular tests in order to directly demonstrate the viral presence in the sample. On the other hand, in cases of peculiar amplification like in this study, metagenomics is a suitable method for viral identification due to the unbiased character of sequence detection. We added the following information in the manuscript:

Lines 245-248, Page 7: “…Nevertheless, viral metagenomics and bioinformatic analysis show a vast and diverse number of challenges that limit their routine application in the clinical practice vary-ing from validation, standardization, cost, turn-around time and complex computer requirements. …”

The methodology described by the authors is insufficient to determine the quality and applicability of the results presented. I strongly recommend to incorporate a quality system around the sequencing workflow and bioinformatics to determine detection limits, sensitivity, specificity, positive predictive value, negative predictive value for the viral targets of interest. Surely, after employing such framework, the impact of these results would increase substantially.

We are grateful for the valuable colocation of Reviewer #2.

We used metagenomic platform that was already optimized for viral identification (Nunes Valença et al., 2020). In brief, we optimized the platform using pools of clinically important viruses at different viral loads that were consequently identified by bioinformatic analysis. We also optimized and parallelized the steps that are used in the bioinformatic analysis. The control of the obtained sequences after this sequence run was performed using bioinformatic analysis where contaminants and artifacts were identified as discussed in the literature i.e. presence of similar virus reads, sequences originating from clones or environmental sources (Asplund et al., 2019)

We agree that next-generation sequencing and bioinformatic analysis have still a long way to go to be routinely applied in the clinical practice due to issues regarding the detection limits (sequences with higher viral load are preferentially detected), specificity and predictive values as well as the presence of contaminants and artifacts. However, the metagenomics is the best method of choice for identification of unknown viruses that are not detected by the conventional PCR techniques due to their limitations This was also the objective of this study, to apply the metagenomics for identification of the viruses by qualitative manner in samples with peculiar PCR amplification profile.

I also strongly recommend extensive editing to improve on the spelling, grammar, and syntax text in the manuscript. The title has a glaring typographical error (Emrging -> Emerging).  

We are sorry for the committed involuntary error in the title. It was corrected accordingly.  

Round 2

Reviewer 1 Report

The authors have replied to all my comments and significantly improved the manuscript.